# Development of a Web-Based Diabetes Prevention Program (DPP) for Chinese Americans: A Formative Evaluation Approach

**DOI:** 10.3390/ijerph20010599

**Published:** 2022-12-29

**Authors:** Ming-Chin Yeh, Wincy Lau, Zoey Gong, Margrethe Horlyck-Romanovsky, Ho-Jui Tung, Lin Zhu, Grace X. Ma, Judith Wylie-Rosett

**Affiliations:** 1Nutrition Program, School of Urban Public Health, Hunter College, City University of New York, New York, NY 10065, USA; 2Department of Health and Nutrition Sciences, Brooklyn College, City University of New York, New York, NY 11210, USA; 3Department of Health Policy and Community Health, Jiann-Ping Hsu College of Public Health, Georgia Southern University, Statesboro, GA 30458, USA; 4Center for Asian Health, Lewis Katz School of Medicine, Temple University, Philadelphia, PA 19140, USA; 5Department of Epidemiology and Population Health, Albert Einstein College of Medicine, Bronx, NY 10461, USA

**Keywords:** Chinese Americans, web-based diabetes prevention program, cultural and linguistic adaptation, focus groups, formative evaluation, qualitative study

## Abstract

Increasing evidence demonstrates that an online Diabetes Prevention Program (DPP) can delay the onset of type 2 diabetes. However, little has been done for Chinese Americans. This study, using Community-Based Participatory Research and Intervention Mapping approaches, describes a formative research process in the development of a culturally and linguistically tailored online DPP program among Chinese Americans with prediabetes living in New York City. Using a triangulation approach, data were collected to inform the development of an online DPP curriculum through (1) a literature review, (2) three focus groups (n = 24), and (3) a community advisory board meeting among 10 key informants knowledgeable in community needs, diabetes care, and lifestyle interventions. Participants indicated online DPPs would be very useful and easily accessible. However, key barriers including low computer skills/literacy and technology self-efficacy were identified. In addition, taking meal photos and tracking pedometer steps daily were found to be acceptable self-motoring tools for sustaining a healthy lifestyle. Furthermore, the integration of features such as text message reminders and the creation of social support groups into the online DPP curriculum was proposed to minimize attrition. This theory-based formative research to develop a culturally and linguistically appropriate web-based DPP curriculum was well-received by Chinese Americans and warrants testing in future intervention studies.

## 1. Introduction

Diabetes prevalence has been growing substantially in the United States. According to the National Diabetes Statistics Report 2020, approximately 34.2 million adults aged 18 and older had diabetes, and 88 million adults had prediabetes [1]. Asian Americans in New York City had the highest prevalence of diabetes in 2014 [2], with half of this population diagnosed as having diabetes or prediabetes despite their relatively lower body weight than other populations [3,4]. The Asian American Federation reported that the Chinese American community accounts for the largest Asian subgroup living in New York City (NYC), with a population increase of 16% from 2010 to 2015 and 70% being foreign-born [5]. An earlier report of a community-based health screening by our research team found that more than one in three (38.3%) Chinese Americans in NYC had impaired fasting glucose (IFG) or diabetes [6].

The Diabetes Prevention Program (DPP) was first introduced in 1996 and demonstrated that intensive lifestyle intervention aiming to achieve a minimum of 7% of weight loss and increased physical activity in individuals with impaired glucose tolerance (IGT), effectively reduced the incidence of type 2 diabetes by 58% compared with a 31% reduction among people who used metformin; furthermore, the risk reduction findings persisted for 10 years [7]. More recently, growing evidence shows that the use of electronic health (eHealth) DPP interventions is as effective as in-person DPP interventions [8]. In addition, the internet penetration rate reached 89% in 2016, and Asian households were most likely to have internet access [9]. Furthermore, elderly populations are seen to have internet access, and 53% of seniors, age 65 years or older, own a smartphone, based on a survey report by the Pew Research Center in 2019 [10]. These statistics suggest the possibility of expanding the reach of web-based DPP in the United States, particularly for formerly overlooked and underserved populations. A systematic review evaluated 26 DPP-based lifestyle interventions delivered through electronic means and indicated that the interventions are effective in weight reduction [11]. A number of authors have recognized the success of web-based DPP in improving clinically meaningful diabetes biomarkers and proved that web-based DPP interventions are cost-effective when compared to a control group [12,13,14], suggesting that eHealth can maintain long-term lifestyle changes with minimal cost and lower the barriers for participants who are less motivated to participate in in-person DPPs.

The National Diabetes Prevention Program (NDPP) was created in 2010 to provide evidence-based lifestyle interventions for preventing type 2 diabetes [15]. Our previous randomized controlled trial (RCT) demonstrated that a culturally and linguistically tailored DPP has a significant impact on reducing Chinese Americans’ diabetes risk in terms of weight reduction and glycemic (HbA1c) improvement compared to the control participants who received general health information in the mail [16]. In view of the promising results of web-based DPPs in delaying the onset of diabetes, our goal was to develop an effective, culturally-appropriate web-based program for the Chinese American community. We involved community stakeholders and key informants to inform all phases of program development and provide community feedback on the web-based program. This article describes the formative research and process of developing a culturally and linguistically tailored web-based DPP intervention for Chinese Americans with prediabetes.

## 2. Methods

The research was conducted in two phases. The first phase involved a formative evaluation which sought available literature and input from study participants and community advisory board (CAB) members, via focus group discussion (FGD), to review the proposed intervention and to suggest ways that would make the intervention easier to implement and sustain. The second phase, incorporating feedback from phase one, was to develop an electronic DPP curriculum and associated materials that are acceptable and feasible for Chinese Americans with prediabetes. The study was approved by the Institutional Review Board of Hunter College, City University of New York (protocol #2018-1008).

### 2.1. Theoretical Framework

The development of the revised DPP for Chinese Americans considered the health planning logic of the PRECEDE-PROCEED Model (PPM) [17]. We adapted the DPP intervention for implementation in a Chinese American population using a social-ecological perspective, which considers environmental factors and social context to promote and support healthy behaviors [17,18]. The PPM framework was used to examine barriers and enablers, especially in the context of the culture and food preferences, of Chinese Americans in adapting the DPP program. In addition, the development of the revised DPP curriculum was guided by Community-Based Participatory Research (CBPR) principles [19]. A CBPR approach leverages community resources and respects cultural sensitivity to build partnerships to achieve mutually agreed-upon outcomes that directly involve community members in all aspects of research and activities [20].

### 2.2. Formative Evaluation Method

We used an Intervention Mapping Approach (IMA) [21] to modify and refine the DPP curriculum and other proposed activities. This approach seeks study participant input to review the proposed intervention and to suggest ways that may make the intervention easier to implement and sustain. Feedback from participants as well as the CAB members was used to refine the curriculum and other related activities. Specifically, the formative evaluation to develop a web-based DPP protocol was based on a triangulation approach [22] using (1) a literature review, (2) a focus group discussion, and (3) community advisory board (CAB) feedback. Figure 1 shows the formative evaluation process using methodological triangulation. An iterative and incremental process was followed in which early feedback was adapted to inform focus group questions, content, and the delivery method of the web-based intervention.

#### 2.2.1. Literature Review

An extensive literature review on web-based diabetes prevention interventions was performed using databases such as PubMed. Information related to dietary habits, physical activity, diabetes management, and the use of technology were searched to inform the development of focus group questions, web-based intervention strategies, and cultural sensitivity. Additionally, facilitators and barriers to a healthy lifestyle (e.g., diet and physical activity) were collected. Cultural and linguistic issues related to immigrants for their health promotion and disease prevention were also collected. A focus group discussion guide was developed based on the collected information. Focus group questions were structured to elicit participants’ feedback on the acceptability of the web-based DPP curriculum, with an emphasis on self-monitoring behavioral changes.

#### 2.2.2. Focus Groups Discussion

Three FGDs consisting of a total of 24 participants were conducted in New York City. Eligible participants were Chinese Americans with prediabetes {HbA1c 39–46 mmol/mol (5.7–6.4%)}, a body mass index (BMI) ≥ (23) kg/m^2^, Chinese-speaking, and aged 18 years and older. Participants were recruited through physician referrals and flyers posted in local community centers. The focus groups were conducted in person between March and April 2019.

Prior to the FGD, participants provided written informed consent. The FGD used an open-ended but semi-structured method so that the conversation and analyses were focused on key questions related to the proposed materials and delivery components. Participants were given five to six of the translated DPP modules used in our previous study [16] and were asked to review the content. Participants were then asked a series of questions based on three major categories: (1) overall reaction to the content of study materials (images, color, fonts, and translations of the curriculum), (2) perceptions and experiences with web-based interventions (smartphone applications were shown to participants), and (3) strategies for maintaining a healthy lifestyle. The FGDs were conducted in Cantonese and/or Mandarin by the study principal investigator and a team member, who are both bilingual and of Chinese heritage. The sessions were audio-taped and lasted approximately one hour.

#### 2.2.3. Community Advisory Board Meeting

To further evaluate the comprehensiveness of the revised curriculum and the acceptability of the web-based intervention, we held a community advisory board meeting with physicians who serve predominantly Chinese Americans and local community leaders who have expertise in diabetes care and lifestyle intervention to solicit feedback on the revised curriculum and the online curriculum. In addition, two Chinese American students with nutrition backgrounds also participated. A total of 10 people participated in the CAB meeting, which was conducted in March 2020. CAB members received the revised curriculum prior to the meeting and shared their opinions and feedback on the cultural appropriateness of the curriculum and proposed activities during the meeting. An online curriculum prototype was also demonstrated onsite to the CAB members to collect feedback on acceptability, feasibility, and other barriers to a web-based intervention. The meeting lasted about two hours, and the audio was recorded.

A thematic analysis was used for the three focus groups and the CAB meeting [23]. Utilizing open coding, two research team members (WL and SC) independently analyzed the data with Atlas.ti (version 8.0) [24], then repeatedly modified the codes together to resolve coding differences. A third person (MCY) was invited to discuss any coding discrepancies. Emerging themes were then generated. Themes were defined by an inductive approach [23]. Principles of reflexivity were also followed and practiced [25]. The authors (MCY, ZG, and WL) who designed the study, translated the curriculum, and collected and analyzed the data for the FDG are all bilingual both in Mandarin and/or Cantonese and are of Chinese heritage with extensive experience in working with the Chinese American community. Considerations such as cultural background, linguistic tradition, personal preferences, and social position were observed, especially during data collection and analysis, to reduce potential bias. For example, a food item may be named differently by people from different regions. The researchers would then discuss and seek agreement for a name that is known and recognized by participants to limit misinterpretation. In addition, aspects of social and cultural theories important to Chinese Americans were used in the interpretation of findings [26]. A detailed description of the process and findings was published in a previous article [27].

## 3. Results

### 3.1. Focus Group and CAB Meeting Finding

Table 1 demonstrates five key themes and twelve sub-themes. The five key themes were (1) Barriers to behavioral changes; (2) Feedback on curriculum content and suggestions; (3) Web-based intervention acceptability; (4) Web-based intervention feasibility; and (5) Web-based intervention implementation and modifications. In addition, a network diagram was developed during thematic analysis to show the interconnectedness of the key themes and sub-themes (see Figure 2). To highlight how we used the focus groups’ contribution to adapt the online DPP curriculum, we will discuss the web-based intervention’s acceptability and feasibility, the use of self-monitoring tools, and web-based intervention implementation and modifications in this article.

#### 3.1.1. Web-Based Intervention Acceptability and Feasibility

The online DPP was well-received: focus group participants expressed that web-based programs would be useful and easily accessible. For example, one participant said that he could use his phone to review the curriculum while on public transportation: *“It is better for (online) sessions like this than to simply read books. In the long-term it is very beneficial.”* However, low computer skills/literacy, and low technology self-efficacy (especially among older adults) were key barriers that participants identified as precluding people from participating in online programs. For example, one common sentiment shared by study participants was, *“I don’t know how to get online.”* The majority of participants said they required instructions for web-based programs due to a lack of online experience.

#### 3.1.2. Acceptability for Web-Based Self-Monitoring Tools

Participants were asked if they were interested in monitoring their diet and physical activities using smartphone applications. While some mentioned they were not familiar with using applications on smartphones, some expressed interest as it was more convenient, and said they were receptive if instructions were provided. Participants also suggested photo sharing on social media as a method to promote diet monitoring as younger generations tend to take photos at mealtimes. Furthermore, the acceptability of using a pedometer to keep track of physical activity was very high among Chinese Americans as they said they thought the pedometer helped increase their physical activity.

#### 3.1.3. Web-Based Intervention Implementation and Modifications

Participants and CAB members suggested the inclusion of more culturally appropriate foods and photos, the provision of videos and short explanations of the activities for better clarity, and formatting and translation of the content in a way that is more familiar to the Chinese American community. Additionally, both participants and CAB members agreed that providing technical support to participants, such as a program manual with step-by-step instructions, would be important and would increase the success of the online DPP program. For example, one participant stated *“If you want to put this (curriculum) online, you have to teach them how to use it. If they don’t know how and you ask them to do it, of course they will say it’s no good.”*

### 3.2. The Use of Formative Evaluation in the Development of an Online DPP Curriculum for Chinese Americans

Using information collected from the literature review, FGD, and CAB, the PreventT2 curriculum from NDPP [28] was adapted into an online 12-month intervention with the intention to create a self-study program that emphasizes self-monitoring. The revised DPP program includes 16 weekly core sessions and 6 of the 10 monthly maintenance sessions. In addition, since the program will be conducted online via Facebook, a user manual was developed with step-by-step instructions on creating a new Facebook account and on completing the weekly/monthly modules and homework. Each participant will be assigned an email address and password to sign up for a Facebook account with a pseudonym. Troubleshooting sessions will also be included to help participants navigate the process of using self-monitoring tools through Qualtrics (Qualtrics, Provo, UT, USA), a web-based survey tool that helps with survey distribution and data collection.

### 3.3. The Online DPP Curriculum

We aimed to design a program that could be easily accessible with the use of smartphones. The online DPP modules were housed in a private Facebook group that is accessible only to study participants and research staff; program modules are delivered as photos in the weekly/monthly post. Figure 3 shows a sample weekly module of the online DPP curriculum in the Chinese language. A typical weekly module includes reading materials and two Qualtrics links for self-monitoring data collection; one for online homework and the other for food intake and a step record logbook.

In order to collect self-monitoring data efficiently, reduce the burden of learning new technology, and enhance participants’ motivation, we streamlined the process of using multiple smartphone apps and developed an electronic survey using Qualtrics. Each week of the core sessions or each month of the post-core sessions, participants would log on to the Facebook private group and read the corresponding module, and then complete homework through two separate Uniform Resource Locator (URL)s which are provided as hyperlinks that will direct them to the Qualtrics survey in the Facebook post together with the module. One is to evaluate the materials learned from the corresponding topic and collect consumer experience data; and the other is to collect their food intake and step records as self-monitoring tools (see Figure 4 and Figure 5), which will be further discussed below. Each online module has 5–15 pages of materials to be reviewed as well as homework to be completed. It would take approximately 15–20 min to complte a module.

#### 3.3.1. Self-Monitoring Tools

The development of DPPs involved the use of behavioral science theories [29]. The self-monitoring of diet and physical activity is an important part of the original DPP as it helps monitor progress and promotes self-efficacy for lifestyle changes. According to the FGD feedback, the majority of participants were interested in tracking their diet and physical activity without the need to download additional applications on their smartphones. Therefore, our goal was to create a simple self-monitoring tool for the participants and the research team to collect data. As mentioned above, a Qualtrics survey was developed that enabled participants to record step counts with a provided digital pedometer and to upload meal photos. Participants were encouraged to report at least three days of step records and dietary intake data weekly to the Qualtrics survey. The data collected will be used to evaluate the feasibility and acceptability of the program.

#### 3.3.2. Social Support

As social support is key to success in diabetes prevention [30], the revised DPP intervention will offer several interactive features. To facilitate social support, participants will be encouraged to reach out to the research team through text messages or phone calls. In addition, the research team will send text message reminders on a weekly basis to remind participants to complete the modules and keep track of their progress. Moreover, there will be an additional Facebook private discussion group for participants to share healthy lifestyle information and act as a support group. These features are designed to help participants stay engaged and prevent them from dropping out of the study.

## 4. Discussion

While DPPs have been translated and implemented in different settings [31,32,33,34,35] and populations, including minorities [32], unique challenges are inherent in adapting the DPP lifestyle intervention for implementation in the Chinese American community. For instance, low-income Chinese and other Asian Americans often live in insular communities with limited exposure to the dominant mainstream culture. In addition, similar to other immigrant groups, Chinese Americans often work long hours with multiple jobs and do not have much free time to participate in a health screening or health promotion events. Finding new strategies, such as delivering health programs online, could potentially overcome many barriers immigrants often face. There are a number of virtual DPPs implemented either online using a desktop computer or with a smartphone-based application that may help increase engagement [13,36,37]. However, to the best of the authors’ knowledge, none has been developed and culturally and linguistically tailored to Chinese Americans.

We used a theory-based, triangulation, and community engagement approach to provide feedback for the development process. It is suggested that employing various approaches in data collection will enhance the validity of the data collected and produce convergent findings of complex health issues [38], and, in our case, develop a comprehensive and innovative online DPP curriculum for Chinese Americans.

Our formative research provides a medium for gathering information and contextualized materials to maximize behavioral change [39]. Based on our focus group findings, the acceptability of the translated and culturally adapted PreventT2 curriculum was high. Many Chinese Americans provided positive feedback and reported they would be interested in and would participate in future online DPPs. They also expressed concern for some barriers (technical complexity and internet access, etc.) that are common to online health programs [27,40].

As Dejoy et al. suggested, the major challenge in translating DPP is to reduce the program cost without sacrificing effectiveness, with a focus on simplicity or ease of implementation [41]. One of the key features in our adaptation was to streamline the number of applications needed for program delivery and data collection. As the participants expressed interest in using web-based tools for self-monitoring, we explored different mobile health technologies that were already on the market and commonly used by dietitians for tracking physical activity and food intake [42,43]. However, the available apps we reviewed did not align with our goals of reducing the learning burden on participants, and participants felt overwhelmed when the apps were shown to them during the focus groups. Therefore, we decided to create a system where we could deliver the program effectively and collect data without compromising the program’s simplicity. Since the use of online questionnaire survey platforms and Facebook for program delivery in many research studies has been emerging [44,45], using Facebook and Qualtrics aligned with our need to tailor and optimize the features in the web-based CDPP. Facebook is not only a well-known social media platform, with which most of the participants were familiar, but also a cost-effective platform for researchers to deliver health programs to hard-to-reach audiences [45,46].

As the CAB members pointed out, individuals had different learning styles and motivation levels, thus, our adaptation also focused on self-study and self-monitoring, and provided assistance as needed. They also suggested training on-site personnel for program implementation. This method was adapted by the Fuel Your Life DPP [41], in which lifestyle coaches were substituted with existing on-site personnel who were trained to answer questions, help with problem-solving, and provide social support to participants. In addition, we included text message reminders, a helpline for participants for technical troubleshooting, and a social support group as these features were shown to provide instrumental and emotional support for participants in other web-based diabetes-management intervention programs [40,47,48]. Prior literature also demonstrated that text messages were a sustainable approach to providing peer support and disseminating health messages in a diabetes self-management program [49].

The use of innovative technologies for dietary assessment has been increasing in research settings [50]. It has been shown that using such approaches was effective in aiding dietary data collection among Asian populations [51]. In recent years, web-based food records have been widely available for self-monitoring and were shown to lower respondent burden [52]. Moreover, studies suggest that image-assisted methods can improve the accuracy of self-reported food records as researchers can estimate portion sizes via the pictures captured by the subjects [53,54]. During our formative research, we developed an electronic form to collect data from participants to monitor their dietary and physical activity behavior (see Figure 4 and Figure 5). The benefits of using this automated data collection system allow the researcher to seamlessly collect data such as photos and increase the completion rate of the questionnaires as participants have the flexibility to complete the surveys on their own schedule. Moreover, it could reduce researchers’ workload and costs [50]. A report by Vandelanotte et al. [55]. identified that underserved populations may be easier to reach using web-based approaches when compared to in-person settings. The use of online social networks and online surveys may increase the feasibility of the intervention as internet access and mobile device ownership further increase with technological advancement.

A major strength of our program was the use of an iterative process and triangulation approach. Although repetitive and time-consuming, it provides a comprehensive method of incorporating the insights gathered in prior literature and feedback from the community in terms of content modifications. In addition, engaging the community and adapting an existing evidence-based curriculum into a digital model was vital. Previous studies have demonstrated that the use of community engagement in formative research was effective in developing a culturally appropriate intervention and building community partnerships [56,57]. Our adaptation model also provides an opportunity for the social marketing of the lifestyle change program in the community [58] which may increase the participation of the program in the future.

Although we were able to develop a low-cost, simplified process and self-monitoring tools in the hope of enhancing the feasibility of the DPP program, there are a few limitations associated with the project. First, the focus group participants were recruited from NYC neighborhoods and may not be representative of participants living in suburban or rural areas where access to cultural foods is not available. Additionally, the majority of participants originated from the southern part of China, and their food preferences and dietary habits may differ compared to those from other parts of China. Further adaptions for different regions may be needed. Moreover, the intervention and measurements relied on self-reported data, which was highly dependent on participants’ engagement and motivation.

## 5. Conclusions

This formative evaluation research was used to develop an easily accessible medium for diabetes prevention in Chinese Americans. We developed a culturally and linguistically appropriate web-based DPP curriculum that could impact health behavioral changes based on the interests of the Chinese American constituents. Although this study protocol is designed to deliver online DPP programs in a low-cost, less resource-intensive way, the need for support is likely to increase in a large-scale rollout. The implementation of the web-based DPP for Chinese Americans and the evaluation of its feasibility, acceptability, and efficacy warrant additional research. In addition, future studies should examine the cost-effectiveness of online DPP programs to facilitate scaled-up implementation and dissemination.

## Figures and Tables

**Figure 1 ijerph-20-00599-f001:**
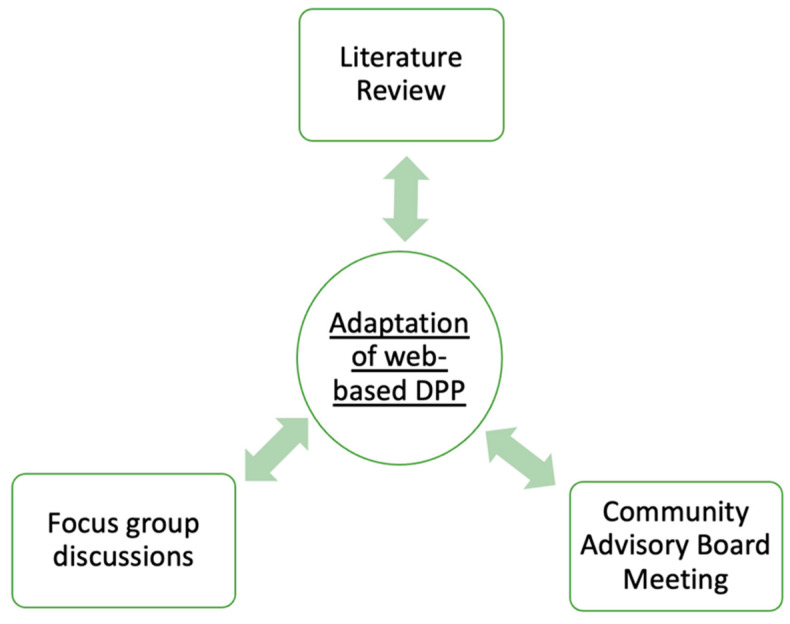
Formative research using methodological triangulation.

**Figure 2 ijerph-20-00599-f002:**
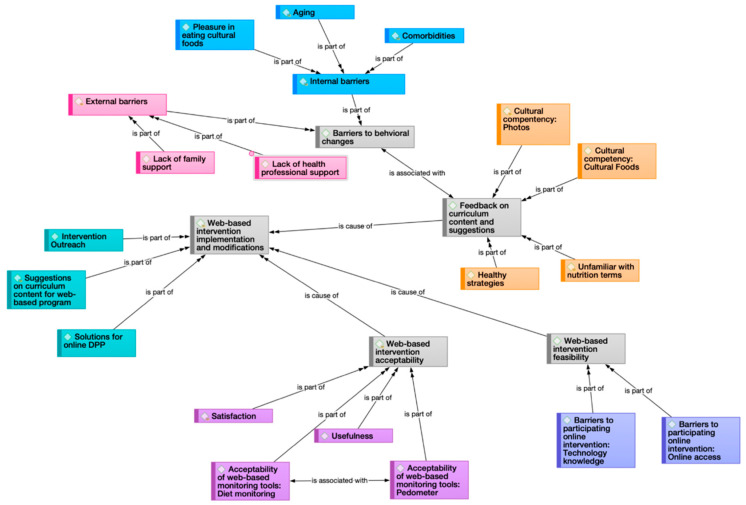
Network diagram showing the relationship among the five major themes.

**Figure 3 ijerph-20-00599-f003:**
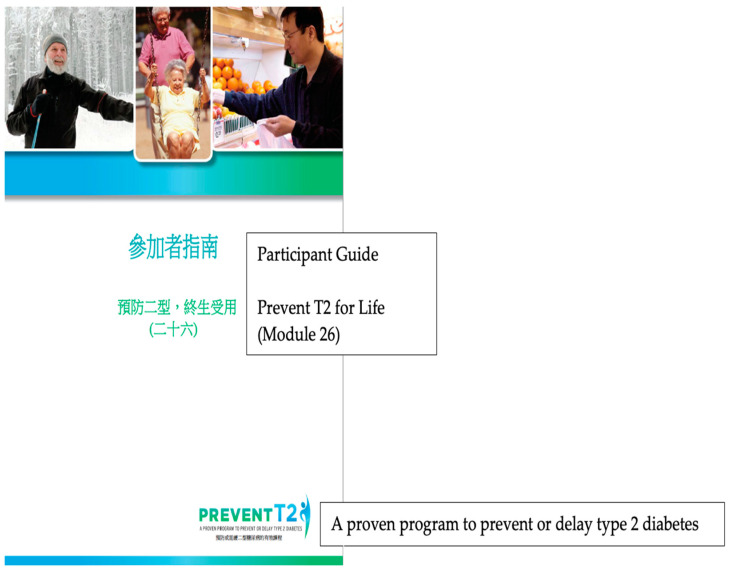
Screenshot of a web-based Chinese diabetes prevention weekly module for adults with prediabetes.

**Figure 4 ijerph-20-00599-f004:**
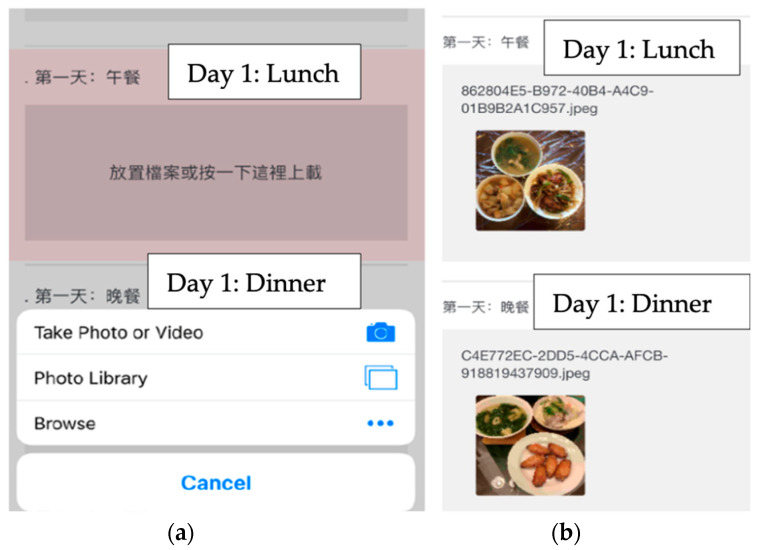
Screenshot for food intake data collection as self-monitoring tools: (**a**) Participants can upload photos from their photo library; (**b**) Example of an uploaded photo for day 1 lunch and dinner.

**Figure 5 ijerph-20-00599-f005:**
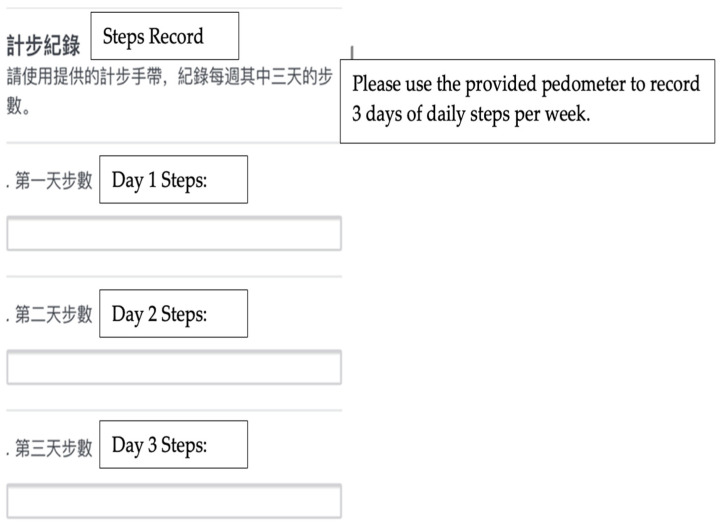
Screenshot for step record data collection as a self-monitoring tool. Participants can upload their three days of step records weekly.

**Table 1 ijerph-20-00599-t001:** Key themes and sub-themes developed during the thematic analysis process.

Key Themes	Sub-Themes
Barriers to behavioral changes	- Internal barriers - External barriers
Feedback on curriculum content and suggestions	- Cultural competency - Unfamiliar with nutrition terms - Healthy strategies
Web-based intervention acceptability	- Satisfaction - Usefulness - Acceptability of web-based monitoring tools
Web-based intervention feasibility	- Barriers to participating in an online intervention
Web-based intervention implementation and modifications	- Suggestions on curriculum content for a web-based program - Intervention outreach - Solution for online DPP curriculum

## Data Availability

The data that support the findings of this study are available from the corresponding author (M-C.Y.) upon reasonable request.

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
