# Peer review of "Development of a Web-Based Diabetes Prevention Program (DPP) for Chinese Americans: A Formative Evaluation Approach"

_ijerph, 2022, doi:10.3390/ijerph20010599_

Round 1
Reviewer 1 Report
This is a well presented description of the development of a Web-based Diabetes Prevention Program (DPP) for Chinese Americans. It addresses a specific important target group. There is excellent co-design involving the key stakeholders. I wont go into detail but the triangulation approach maximises this.
It would be helpful if the authors could provide additional information most likely in the discussion section.
The Chinese community is heterogeneous and the existence of Cantonese and Mandarin speaking communities is already mentioned. Country of origin, length of stay in USA, age of participants are likely to be relevant. in taking photos of meals in some cases these are shared and this impacts on portion sizes. Did the focus group participants think they could stick with the program for 12 months? In other DPPs attrition over time is a major issue. How feasible is large scale roll out as the needs for support escalate?
Author Response
The authors would like to thank the reviewers for their thoughtful comments and suggestions. Our responses to reviewers’ comments are provided below.
This is a well presented description of the development of a Web-based Diabetes Prevention Program (DPP) for Chinese Americans. It addresses a specific important target group. There is excellent co-design involving the key stakeholders. I wont go into detail but the triangulation approach maximises this.
R: Thank you!
It would be helpful if the authors could provide additional information most likely in the discussion section.
The Chinese community is heterogeneous and the existence of Cantonese and Mandarin speaking communities is already mentioned. Country of origin, length of stay in USA, age of participants are likely to be relevant. in taking photos of meals in some cases these are shared and this impacts on portion sizes. Did the focus group participants think they could stick with the program for 12 months? In other DPPs attrition over time is a major issue. How feasible is large scale roll out as the needs for support escalate?
R: Thank you for the reviewer’s thoughtful comments and questions. During the focus group interviews, we did not ask specifically whether participants would be able to stick with the program for 12 months. However, based on the finding from our previous study (Yeh M-C, et al. Translation of the diabetes prevention program for diabetes risk reduction in Chinese immigrants in New York City. Diabet Med. 2016;33(4):547-551. doi:10.1111/dme.12848), we observed a less than 5% attrition at the end of 12 months. It was a small pilot study with only 60 participants. It is possible that attrition will be higher in a larger study.
The current study protocol is designed to deliver DPP program online in a low cost, less resource intensive way. The feasibility of a large-scale roll out is high. Nonetheless, as the reviewer rightfully pointed out, the needs for support will likely increase in a large study. We have added the following sentence in the conclusions to highlight this important point: “Although this study protocol is designed to deliver online DPP programs in a low cost, less resource intensive way, the needs for support are likely to increase in large-scale roll out.” Please see lines 363-5.
Reviewer 2 Report
Dear authors, a significant and translational study within the diabetes prevention research space.
Context and methodology are described well. The topic is relevant and this addresses a specific gap.
Results section could improve. I would suggest to add more content from focus group discussion. Currently this section lack privileging subjective meanings. Adding further quotes from participants that reflects identified themes and subthemes would be helpful. Results section requires more content from Focus Group discussion.
Researcher reflexivity is said to be accounted however how this was done is not described - please add a sentence or two regarding the same in Methods.
What is the next step after implementing the web-based DPP? Adding one or two sentences about future work will be useful.
Author Response
The authors would like to thank the reviewers for their thoughtful comments and suggestions. Our responses to reviewers' comments are provided below.
Dear authors, a significant and translational study within the diabetes prevention research space.
Context and methodology are described well. The topic is relevant and this addresses a specific gap.
R: Thank you!
Results section could improve. I would suggest to add more content from focus group discussion. Currently this section lack privileging subjective meanings. Adding further quotes from participants that reflects identified themes and subthemes would be helpful. Results section requires more content from Focus Group discussion.
R: thank you for the reviewer’s comments on the need to expand the focus group findings. The authors just published a paper recently detailing the full report of analysis and summary of those focus group findings (Yeh M-C, Lau W, Chen S, et al. Adaptation of diabetes prevention program for Chinese Americans – a qualitative study. BMC Public Health. 2022;22(1):1325. doi:10.1186/S12889-022-13733-5). The authors agree that focus group findings are an important and integral part of the study. Due to space limits, regrettably, we were able to include only the points on web-based intervention acceptability, implementation and modifications, and feasibility and acceptability for self-monitoring tools that are most salient to the current manuscript.
Researcher reflexivity is said to be accounted however how this was done is not described - please add a sentence or two regarding the same in Methods.
R: Thank you for this important comment. We have added the following sentences to the methods section: “Principles of reflexivity were also followed and practiced [26]. Authors (MCY, ZG, WL) who designed the study, translated the curriculum, collected and analyzed the data for the FDG are all bilingual both in Mandarin and/or Cantonese and of Chinese heritage with extensive experience in working with Chinese American community. Considerations such as cultural background, linguistic tradition, personal preferences, and social position were observed, especially during data collection and analysis, to reduce potential bias. For example, a food item may be named differently by people from different regions. The researchers would discuss and seek agreement for a name that is known and recognized by participants to limit misinterpretation." Please see lines 165-173.
What is the next step after implementing the web-based DPP? Adding one or two sentences about future work will be useful.
R: The authors agree with the reviewer’s comment fully. We have added the following sentence to the conclusions: “In addition, future studies should examine the cost-effectiveness of online DPP programs to facilitate scale-up implementation and dissemination.” Lines 467-9.